# Resolved imaging confirms a radiation belt around an ultracool dwarf

Melodie M. Kao[1,2,5] ✉, Amy J. Mioduszewski[3,5], Jackie Villadsen[4] & Evgenya L. Shkolnik[2]

Radiation belts are present in all large-scale Solar System planetary magnetospheres: Earth, Jupiter, Saturn, Uranus and Neptune[1]. These persistent equatorial zones of relativistic particles up to tens of megaelectron volts in energy can extend further than ten times the planet's radius, emit gradually varying radio emissions[2–4] and affect the surface chemistry of close-in moons[5]. Recent observations demonstrate that very low-mass stars and brown dwarfs, collectively known as ultracool dwarfs, can produce planet-like radio emissions such as periodically bursting aurorae[6–8] from large-scale magnetospheric currents[9–11]. They also exhibit slowly varying quiescent radio emissions[7,12,13] hypothesized to trace low-level coronal flaring[14,15] despite departing from empirical multiwavelength flare relationships[8,15]. Here we present high-resolution imaging of the ultracool dwarf LSR J1835 + 3259 at 8.4 GHz, demonstrating that its quiescent radio emission is spatially resolved and traces a double-lobed and axisymmetrical structure that is similar in morphology to the Jovian radiation belts. Up to 18 ultracool dwarf radii separate the two lobes, which are stably present in three observations spanning more than one year. For plasma confined by the magnetic dipole of LSR J1835 + 3259, we estimate 15 MeV electron energies, consistent with Jupiter's radiation belts[4]. Our results confirm recent predictions of radiation belts at both ends of the stellar mass sequence[8,16–19] and support broader re-examination of rotating magnetic dipoles in producing non-thermal quiescent radio emissions from brown dwarfs[7], fully convective M dwarfs[20] and massive stars[18,21].

At 77.28 ± 10.34 Jupiter masses[22], the nearby[23] (5.6875 ± 0.00292 pc) M8.5 spectral type[24] ultracool dwarf (UCD) LSR J1835 + 3259 straddles the hydrogen burning mass limit differentiating between low-mass stars and massive brown dwarfs. It is nearly the size of Jupiter with a radius $R_{UCD}$ = 1.07 ± 0.05 Jupiter radii[22] and is edge-on relative to our line of sight with a rotation axis inclined at an angle $i \approx 90°$ (Extended Data Table 2). LSR J1835 + 3259 emits rotationally periodic and bursting[25] 8.4 GHz radio aurorae every 2.84 ± 0.01 h that trace magnetic fields greater than or equal to 3 kG near its surface[6]. It also produces quiescent radio emission at the same observing frequencies[14,25,26] and faint 97.5 GHz emission[27]. The latter is unlikely to be disk or flare emission (Methods) and may instead trace the same population of relativistic electrons as its 8.4 GHz quiescent emission.

Using the High Sensitivity Array (HSA) of 39 radio dishes spanning from the USA to Germany, we searched for extended quiescent radio emission at 8.4 GHz from LSR J1835 + 3259 indicating a stable and large-scale plasma structure as evidence of an extrasolar analogue to Jupiter's radiation belts. Our observing campaign consisted of three five-hour epochs from 2019 to 2020 (Table 1), capturing nearly two full rotation periods per epoch.

We find that quiescent radio emissions from LSR J1835 + 3259 persist throughout each epoch and exhibit a double-lobed morphology that is stable for more than one year (Fig. 1). Up to 18.47 ± 1.85 $R_{UCD}$ separate

its radio lobes, which have no detectable circular polarization in any epoch (Table 1 and Extended Data Table 2). These data reveal resolved imaging of plasma captured in the magnetosphere of a planet-sized object outside our Solar System.

Auroral bursts appear centrally located between the two lobes in Epoch 2, which has the highest-quality data (Figs. 1b and 2). In Epoch 1, missing antennas (Table 1) significantly reduce sensitivity on shorter baselines relevant for detecting extended emission, which causes the lobes to appear more compact and the aurora to coincide with the west lobe (Fig. 1a). We simulate an observation of the Epoch 2 model image using the Epoch 1 antenna configuration and find that Epoch 1 is consistent with this simulated Epoch 2 observation (Extended Data Fig. 1). In Epoch 3 (Fig. 1c), aurorae are too faint to confidently locate, but radio lobe separations are consistent with Epoch 2.

The 8.4 GHz aurorae originate in 3 kG magnetic fields near the surface of LSR J1835 + 3259 (ref. 25). From Epochs 2 and 3, we infer that lobe centroids sit at approximately 9 $R_{UCD}$ from the ultracool dwarf (with 7–10% uncertainties; Table 1), while their outer extents reach at least 12–14 $R_{UCD}$. The structure may be even larger; individual epochs may not be sensitive to fainter and more extended emission, as is the case for Epoch 1.

At these large extents, dipole magnetic fields decaying with radius as $B \propto r^{-3}$ will dominate more rapidly decaying higher-order magnetic

[1]Department of Astronomy & Astrophysics, University of California, Santa Cruz, CA, USA. [2]School of Earth & Space Exploration, Arizona State University, Tempe, AZ, USA. [3]National Radio Astronomy Observatory, Socorro, NM, USA. [4]Department of Physics & Astronomy, Bucknell University, Lewisburg, PA, USA. [5]These authors contributed equally: Melodie M. Kao and Amy J. Mioduszewski. ✉e-mail: mmkao@ucsc.edu

## Table 1 | Position and spatial extent of quiescent emission from LSR J1835+3259

| Epoch: date | Missing antennas[a] | Synthesized beam | Centroid separation | | E diameter[c] | | W diameter[c] | |
|---|---|---|---|---|---|---|---|---|
| | | (mas×mas) | (mas) | ($R_{UCD}$[b]) | (mas) | ($R_{UCD}$[b]) | (mas) | ($R_{UCD}$[b]) |
| 1: 15 June 2019 | VLBA:2, MK, SC, GBT | 2.10×0.43 | 1.04±0.05 | 11.52±0.81 | — | — | — | — |
| 2: 20 August 2020 | MK-EB | 1.71×0.58 | 1.61±0.10 | 17.95±1.39 | 0.58±0.10 | 6.44±1.19 | 0.71±0.12 | 7.87±1.38 |
| 3: 28 August 2020 | MK | 1.61×0.60 | 1.66±0.15 | 18.47±1.85 | 0.66±0.11 | 7.39±1.31 | 0.83±0.19 | 9.28±2.20 |

Each five-hour epoch combines the Very Long Baseline Array (VLBA), Karl G. Jansky Very Large Array (VLA), Green Bank Telescope (GBT) and Effelsberg Telescope (EB). [a]Numbers denote hours if missing for a portion of the observation and a hyphenated pair denotes a baseline. MK, VLBA Mauna Kea; SC, VLBA Saint Croix. [b]$R_{UCD}$=1.07±0.05 Jupiter radii (ref. 22). Radius uncertainties are propagated in reported dimensions. [c]Lobe diameters and errors were obtained by fitting two freely floating elliptical Gaussians to each epoch image. Fitted minor axes for each lobe were resolved except for in Epoch 1 owing to missing antennas. Major axes for each lobe were unresolved in fits.

fields of similar surface field strengths inferred for LSR J1835 + 3259 with multiwavelength spectra[28,29]. Indeed, the persistent double-lobed and axisymmetrical morphology observed is consistent with a stable dipole magnetic field and theoretical treatments assuming such can explain radio aurorae observed from LSR J1835 + 3259 and other ultra-cool dwarfs[9–11].

Our observations present self-consistent evidence for an analogue of planetary radiation belts outside our Solar System, consisting of long-lived relativistic electron populations confined in a global magnetic dipole field[30]. The double-lobed and axisymmetrical geometry observed from LSR J1835 + 3259's quiescent radio emission is similar to the radio morphology of Jupiter's radiation belts[2] and consistent with a belt-like structure about the magnetic equator for this edge-on system (Figs. 1 and 2).

To explore implications of the lobe separation for electron energies, we consider a surface dipole field greater than or equal to 3 kG. For lobe centroids at the magnetic equator, the field strength and corresponding non-relativistic electron cyclotron frequency[31] is 2 G and $v_c \cong 6$ MHz. An electron gyrating about a magnetic field emits at an observed frequency $s(v_c/\gamma)$ that is multiple harmonics $s$ of its relativistic cyclotron frequency, where $\gamma > 1$ is the Lorentz factor of the electron described by its speed[31]. Emission at 8.4 GHz corresponds to $s \geq 1,500$ in the lobe centroids, which rules out gyrosynchrotron emission ($s \approx 10–100$) from

mildly relativistic electrons[31]. More rapidly decaying higher-order magnetic fields result in higher harmonics.

Instead, such high harmonics indicate synchrotron emission from very relativistic electrons, which cannot produce strong circular polarization. Indeed, we do not detect circular polarization in its resolved radio lobes in any epoch (Extended Data Table 2 and Methods). For the less resolved and brighter quiescent emission in Epoch 1, our noise floor gives a 95% confidence upper limit of greater than or equal to 8.8% and 15.5% circular polarization in the east and west lobes, respectively.

Synchrotron emission, instead, produces linear polarization[31], which has been observed at the 20% level for the Jovian radiation belts[3]. Our observations do not include linear polarization calibrations, and so call for future such measurements to confirm synchrotron emission from LSR J1835 + 3259.

We can estimate electron energies from synchrotron emission because each electron emits most of its power near its critical frequency[31] $v_{crit} \approx (3/2)\gamma^2 v_c \sin \alpha$ for pitch angle $\alpha$. For $v_{crit} \approx 8.4$ GHz, electrons with nearly perpendicular pitch angles will have $\gamma \approx 30$. These high Lorentz factors correspond to 15 MeV and are comparable to Jovian radiation belt electron energies up to tens of mega-electron volts[3,4]. Jupiter's GHz radiation belts trace higher energy electrons and are more compact than its 127 MHz radiation belts[2,3,32]. Similarly, our measured 8.4 GHz lobe separations for LSR J1835 + 3259

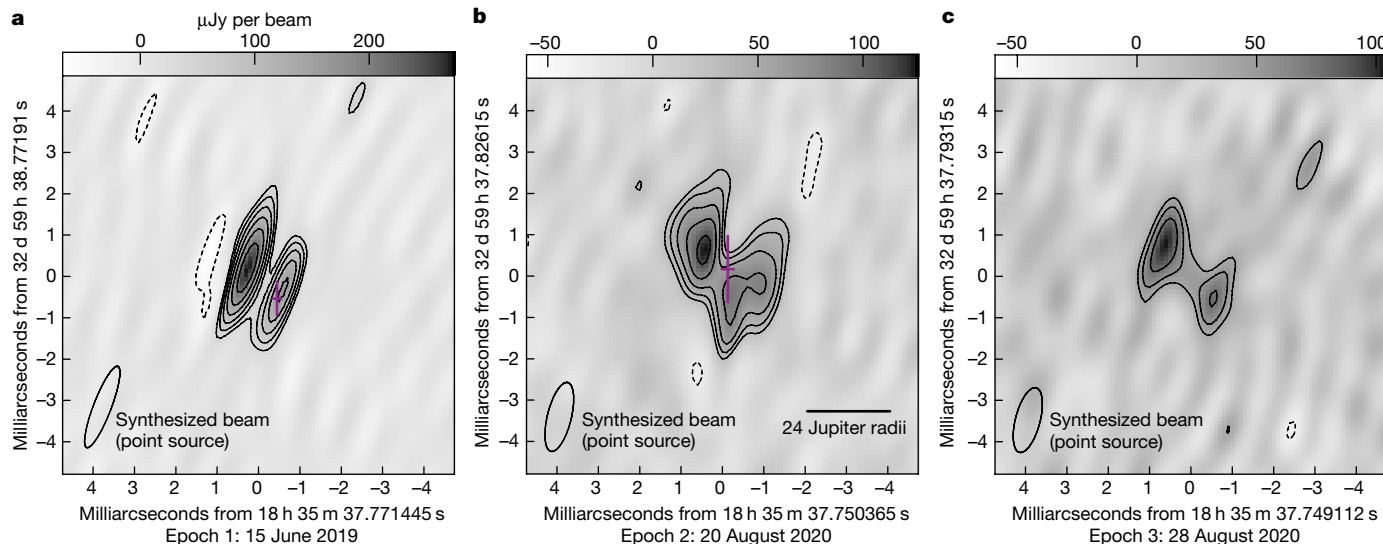

**Fig. 1 | Quiescent 8.4 GHz emission from LSR J1835 + 3259. a–c**, This is resolved in each five-hour epoch on 15 June 2019 (**a**), 20 August 2020 (**b**) and 28 August 2020 (**c**). The synthesized beam sets the resolution size for each image and appears shortened along one axis due to the array configuration. Contours denote $3\sigma_{r.m.s.} \times (-1, 1, \sqrt{2}, 2, 2\sqrt{2}, 4)$ increments, where the root-mean-square deviations ($\sigma_{r.m.s.}$) are given in Extended Data Table 2. Crosshairs indicate aurorae centroids and their $3\sigma$ positional errors (magenta). Coordinates are for midnight in International Atomic Time and east corresponds to the direction of increasing right ascension.

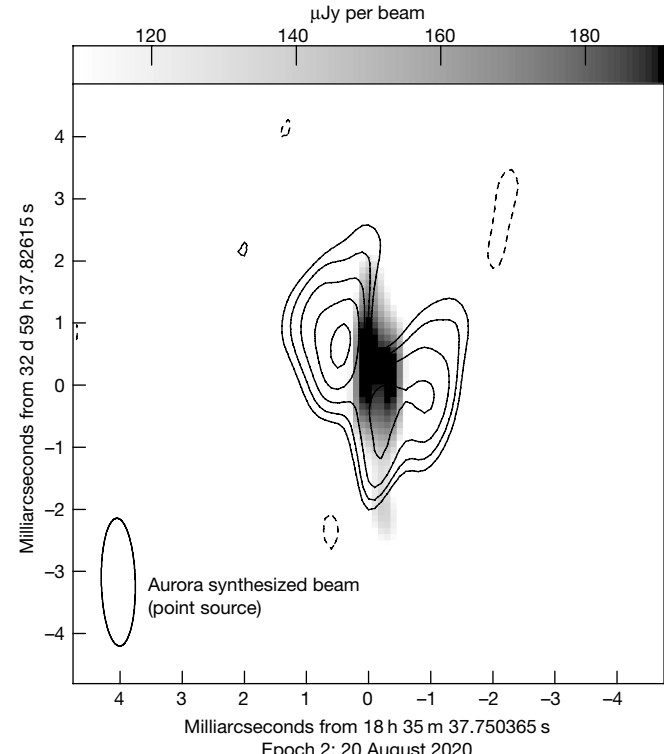

**Fig. 2 | Epoch 2 composite image of LSR J1835 + 3259: aurora and quiescent emission.** The right-circularly polarized aurora was separately imaged and overlaid in grey scale on quiescent 8.4 GHz emission contours from Fig. 1b corresponding to $3\sigma_{r.m.s.} \times (-1, 1, \sqrt{2}, 2, 2\sqrt{2}, 4)$, where $\sigma_{r.m.s.}$ is given in Extended Data Table 2. The synthesized beam sets the resolution element for the aurora and is determined by the array configuration. Figure 1b shows the synthesized beam for quiescent emission. The aurora appears centrally located with respect to the double-lobed morphology of the quiescent emission. Coordinates are for midnight in International Atomic Time.

are approximately 70% more compact than those measured contemporaneously at 4.5 GHz (ref. 33).

Electrons lose energy as they emit synchrotron radiation[31]. We estimate a cooling time $\tau \approx 60$ days for LSR J1835 + 3259 (Methods), yet its double-lobed structure persists for over a year. Although unresolved, quiescent emission at the same observing frequencies has been present for over a decade[14,25,26]. In the standard model of stellar flares[34], photospheric motions build magnetic energy that reconnection events impulsively release, accelerating radio-emitting electrons in the stellar corona[31]. Indeed, during an 8.4 GHz flare, the flare star UV Ceti (M5.5 spectral type) temporarily displayed a double-lobed structure with a cooling time of approximately 2 h separated by 4–5 stellar radii along its inferred rotation axis[35].

Despite a long synchrotron cooling time estimated for LSR J1835 + 3259, stellar flare rates alone cannot explain its persistent radio emission. Ultracool dwarfs of similar spectral type, such as TRAPPIST-1, can optically flare approximately once per day to once per month depending on flare energies[36,37]. For TRAPPIST-1, flare activity produces X-ray emission from coronal heating[38] but not detectable radio emission. Such behaviour agrees with empirical flare correlations[39]. In contrast, the X-ray upper limit for LSR J1835 + 3259 is less than 1% of TRAPPIST-1's quiescent luminosity[14,15], which indicates minimal heating from both instantaneous and time-averaged flare activity (Methods). Subsiding flare activity as objects approach cooler temperatures[36,37] further exacerbates this issue across the ultracool dwarf mass spectrum. Planets do not flare like stars, yet a 12.7 ± 1.0 Jupiter mass brown dwarf[40] straddling the planetary mass limit also exhibits quiescent radio emission[7,41].

Radiation belts around Solar System planets offer alternative acceleration mechanisms and a compelling analogy for interpreting LSR J1835 + 3259's double-lobed quiescent radio emission. In contrast to stellar flares, centrifugally outflowing plasma accelerates while maintaining corotation with Jupiter's magnetosphere[4], stretching and triggering reconnection in its global magnetic field at large distances[42]. Rotationally driven currents spanning Jupiter's magnetosphere[4] and powering its main aurora[43] can also accelerate radiation belt electrons. These processes effectively tap Jupiter's large reservoir of rotational energy. Finally, to reach observed MeV energies, electrons undergo adiabatic heating as they encounter stronger magnetic fields during slow inward radial diffusion[4].

Intriguingly, recent radiation belt modelling for magnetized massive stars also reproduces 8.4 GHz quiescent radio luminosities from LSR J1835 + 3259 (ref. 18). This model ties stellar rotational energy to quiescent radio luminosities[18] and proposes radiation belts heated by a mechanism analogous[21] to rotationally driven reconfigurations of Jupiter's global magnetic field[42]. It explains correlated quiescent radio and Balmer emission luminosities in magnetized massive stars[44]. Ultracool dwarf quiescent radio luminosities also correlate with Balmer emission[7,8] that is interpreted as tracing auroral rather than the usual chromospheric activity[6–8]. This suggests that conditions enabling their aurorae, such as rapidly rotating dipolar magnetic fields[9–11], may support strong quiescent radio emission[8,18].

As such, LSR J1835 + 3259's double-lobed synchrotron emission exhibits properties consistent with an extrasolar analogue to the Jovian radiation belts.

Open questions remain, including the source of ultracool dwarf radiation belt plasma. Ongoing searches for their predicted planets and moons[45,46] may help to show that volcanism from such companions seed ultracool dwarf magnetospheres in a manner similar to Io in Jupiter's magnetosphere[47]. Additionally, unlike Jupiter, stellar-like flares on ultracool dwarfs[36,37] may provide electrons that are later accelerated to the high energies inferred. Variability on days-long timescales observed for LSR J1835 + 3259 (Extended Data Table 2) is also observed from radiation belts around Jupiter and Saturn. For the latter, it is attributed to changes in radial diffusion tied to solar weather[48,49]. We postulate that flaring and/or centrifugal breakout activity may similarly perturb particle acceleration mechanisms in ultracool dwarf radiation belts while augmenting their electron populations.

Beginning with the discovery of ultracool dwarf radio emission[50] and the later confirmation of aurorae occurring on ultracool dwarfs[6], our result completes a paradigm in which planetary-type radio emissions emerge at the bottom of the stellar sequence as stellar-like flaring activity subsides

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

## Methods

### Target parameters

Absorption line modelling for LSR J1835 + 3259 gives a 2,800 ± 30 K effective temperature[28] corresponding to a young age, brown dwarf mass and inflated radius (22 ± 4 Myr, 55 ± 4 Jupiter masses $M_J$, 2.1 ± 0.1 Jupiter radii $R_J$; ref. 28). However, this temperature is inconsistent with the 2,316 ± 51 K expected[22] for its M8.5 spectral type[24] and may be subject to systematic effects in the model atmospheric spectra. Indeed, the young inferred age does not exceed typical M dwarf disk dissipation timescales[51], yet LSR J1835 + 3259 does not exhibit infrared excess indicative of a disk[52] and its periodic Balmer line emission is associated with aurorae[6] rather than accretion. Furthermore, LSR J1835 + 3259 does not have detectable lithium absorption in its atmosphere[53], which indicates that its mass is likely to be higher than the approximately 65 $M_J$ threshold above which lithium depletion occurs and that its age is older than the depletion timescale[54,55]. Instead, the properties we adopt (at least 500 Myr, 77.28 ± 10.34 $M_J$, 1.07 ± 0.05 $R_J$; ref. 22) are consistent with these multiwavelength observations of LSR J1835 + 3259. Extended Data Table 1 summarizes all properties for LSR J1835 + 3259 that we use in our analysis and discussion.

### Observations

The HSA combines the Very Large Baseline Array (VLBA, ten 25-m dishes), the Karl G. Jansky Very Large Array (VLA, twenty-seven 25-m dishes) as a phased array, the Robert C. Byrd Green Bank Telescope (GBT, single 100-m dish) and the Effelsberg Telescope (EB, single 100-m dish). Not all epochs successfully included all telescopes because of weather, equipment failures and site closures. LSR J1835 + 3259 was visible on the longest baseline from the VLBA dish at Mauna Kea, Hawaii (MK) to EB in Bad Münstereifel, Germany (10,328 km) for no more than one hour per observation. We prioritized time on EB to increase observational sensitivity and long baselines to the telescopes on continental USA. Table 1 summarizes presented HSA observations.

To incorporate the VLA in a very long baseline interferometry observation, all antennas in the array must be phase-corrected and summed (that is, phased). The phased-VLA then operates as a single element in the HSA array with a primary beam equal to the synthesized beam of the VLA. This phased-VLA data can also be used as a regular VLA observation.

We observed in A and B configurations for the VLA, giving phased-VLA primary beams of approximately 0.2″ and 0.6″ (half-power beam width), respectively, at our X band (8.4 GHz) observing frequency. To obtain the full sensitivity of the phased-VLA observations, LSR J1835 + 3259 must be within half of the phased-VLA primary beam from the centre of the field.

This can pose a challenge for an object with as high proper motion and parallax as LSR J1835 + 3259 (Extended Data Table 1) and an unexpected positional offset from its Gaia motion-evolved coordinates[23,56,57] (Methods, Target position). To ensure target capture, a week to a month before each HSA observation, we obtained a 60-min observation using only the VLA in array mode at X band to image and locate the position of the target.

We observed the VLBA standard phase calibrator J1835 + 3241 to calibrate phase errors from atmospheric fluctuations. This phase calibrator is within 0.33° of our target and is an International Celestial Reference Frame (ICRF) calibrator source. Our phase calibration cycle periods were 4 min and 8 min for HSA and VLA-only observing blocks, respectively. During HSA observing, we also phased the VLA every 10 min with this same phase calibrator to maintain coherence across the VLA and observed J1848 + 3219 approximately every 2 h for fringe finding and as a check source. Finally, we observed 3C286 as a flux calibrator for the VLA in each epoch to allow independent analyses of the phased-VLA data.

### Calibrations

For HSA observations, we applied standard phase reference very long baseline interferometry data reduction methods[58] using the Astronomical Image Processing System (AIPS) package by the National Radio Astronomy Observatory[59]. The target is too faint to self-calibrate, so it was phase-referenced to the nearby calibrator J1835 + 3241.

The narrow 256 MHz bandwidth of the HSA observations and 8.4 GHz centre frequency largely avoids radio frequency interference as a source of noise. Nevertheless, we carefully examined the data from each epoch to identify and remove bad data. No data from MK was collected for Epoch 1 and calibrating the MK-EB baseline in Epochs 2 and 3 proved very difficult. This was because both telescopes overlapped for only a short time and they created a single very long baseline that was not similar to any other. For these reasons, we excluded the MK-EB baseline in Epoch 2 and MK entirely in Epoch 3 for final imaging.

For each epoch, we calculated the combined apparent motion of our target from both proper motion and parallax using Gaia[23,56] Data Release 3. For the latter, we assumed a geocentric observer on a circular 1 AU orbit. These assumptions give parallax offsets that are within 2% of the true parallax amplitude[60] and well within our resolving power (Table 1). The respective east–west and north–south motions are −0.13 mas h⁻¹ and −0.06 mas h⁻¹ (Epoch 1), −0.09 mas h⁻¹ and −0.17 mas h⁻¹, (Epoch 2) and −0.07 mas h⁻¹ and −0.18 mas h⁻¹ (Epoch 3). Our five-hour observations can resolve this apparent motion, which we correct for using the AIPS task 'clcor'. As a check, we also imaged the data from each epoch without correcting for either proper motion or parallax. We find the same overall double-lobed structure in each epoch but with lower signal-to-noise, as would be expected from the motion smearing effect.

Circular polarization, which is the difference in the right- and left-circularly polarized data, can distinguish between and characterize electron cyclotron maser (approximately 100% circular polarization), gyrosynchrotron (up to tens of percent) and synchrotron emissions (minimal)[31]. Since all telescopes in the HSA use circularly polarized feeds, it is easy for even slightly incorrect amplitude calibration to produce spurious instrumental circular polarization. To ensure that any circular polarization detected was from LSR J1835 + 3259 rather than errors in the calibration, we checked the circular polarizations of our calibrators. These showed instrumental contamination resulting in approximately 7–10% spurious circular polarization in the HSA observations. We separately inspected the VLBA-only and VLA-only data on the phase calibrator and found that these data contained approximately 0.1% circular polarization from instrumental contamination and/or circular polarization intrinsic to the calibrator. To correct the amplitude calibration on the HSA, we self-calibrated[61] the phase calibrator in each epoch (Extended Data Fig. 2) to reduce instrumental polarization to less than or equal to 1%. We then transferred the resulting amplitude calibrations to our target to reduce any instrumental circular polarization to the approximately 1% level.

### Time series

Radio aurorae on LSR J1835 + 3259 manifest as bright, periodic and strongly circularly polarized electron cyclotron maser bursts every 2.84 ± 0.01 h (ref. 25) that are clearly evident even in the stand-alone phased-VLA data. LSR J1835 + 3259 is unresolved in phased-VLA data, for which we produced time series (Extended Data Fig. 3) with the AIPS task 'dftpl' that is specifically designed for unresolved objects. We find that two auroral bursts were partially or fully detected in each epoch for both right- and left-circularly polarized data.

### Quiescent emission imaging

To create images of the quiescent emission, we removed auroral bursts identified in the time series (Extended Data Fig. 3) and imaged the remaining data in each epoch (Fig. 1). All images presented in this article

were imaged using the CLEAN algorithm implemented by the AIPS task 'imgr'. A Briggs robust weighting[62] of 0.0 balances between uniform and natural weighting to allow both high resolution and sensitivity to non-point sources and a 0.1 mas pixel size gives 4–6 pixels across the narrowest part of the synthesized beam.

We observe a double-lobed morphology in each epoch (Fig. 1). Detailed modelling of the quiescent emission to distinguish between different morphology types is outside the scope of this article. Instead, we measure quiescent emission source sizes and flux densities using the AIPS task 'jmfit'. In each epoch image, we fit[63] two elliptical Gaussians with freely floating centres, sizes, peaks and integrated flux densities (Table 1 and Extended Data Table 2). The lobes are resolved along approximately the east–west axis in Epochs 2 and 3. Measured integrated flux densities are consistent with those reported in the literature[14,25,50].

To help to distinguish between synchrotron and gyrosynchrotron emissions, we imaged the total circular polarization (Stokes V; right minus left-circular polarization) for the quiescent emission from LSR J1835 + 3259 in each epoch. There was no detectable circular polarization above the 12–13 μJy beam$^{-1}$ noise floors of the Stokes V images (Extended Data Table 2). As a further check, we also imaged our target using data from the VLBA-only and stand-alone phased-VLA. We find no convincing Stokes V emission to root-mean-square (r.m.s.) noise floors of the 37 μJy beam$^{-1}$ and 30 μJy beam$^{-1}$. These non-detections are consistent with low integrated circular polarization (approximately 8 ± 2%) measured in a previous 11-h VLA observation at 8.44 GHz that also averages over circularly polarized but periodically bursting aurorae[25]. For our brightest quiescent lobe, this previously measured level of circular polarization would be a less than or equal to a 2$\sigma$ source in our HSA Stokes V images.

### Auroral bursts imaging

We imaged auroral bursts in the same way as described in the previous section. However, first we removed the quiescent emission. After obtaining a model of the quiescent emission in each epoch from the Fig. 1 images, we subtracted that model from its corresponding epoch's full and quiescent-only datasets using the AIPS task 'uvsub'. We then re-imaged the latter after subtraction to ensure that no flux remained.

Right- and left-circularly polarized auroral bursts overlap (Extended Data Fig. 3), which can suppress the Stokes V emission. As such, we separately imaged right- and left-circularly polarized emission.

For right-circularly polarized auroral bursts, we imaged only the brightest 15–20 min noted in Extended Data Fig. 3a,b for Epochs 1 and 2. These shorter time ranges avoided averaging longer periods of data with very different flux densities, which can cause artifacts in interferometric imaging. As a check, we also image the Stokes V data and confirm that aurorae are circularly polarized.

In Epoch 3, auroral bursts were too faint to confidently image. A set of extended calibration scans coincided with one of the bursts (Extended Data Fig. 3c). As a result, no data were obtained on-target during its peak brightness, which suppressed the contrast between auroral flux density and imaging r.m.s. noise. We also attempted to image left-circularly polarized auroral bursts, but these were too faint to be imaged for all epochs.

For imaged auroral bursts, we measured their spatial extent, location and flux densities (Extended Data Table 2) by fitting an elliptical Gaussian using the AIPS task 'jmfit'. In Epoch 1, the auroral burst was unresolved. In Epoch 2, the auroral burst was morphologically distinct from the quiescent radio lobes (Fig. 2) and consistent with both being unresolved or marginally resolved along approximately the east–west direction. It has a minor axis of approximately 0.4 mas compared with a 2.06 × 0.55 mas synthesized beam (Extended Data Table 2).

Even with our close-proximity phase calibrator, phase errors likely remain. These can introduce spurious substructure on length scales smaller than the synthesized beam for transient emission. Consequently, we cannot conclude whether the aurora has any physical substructure. This effect averages out over time as phase errors vary and is important only for assessing marginally resolved emission. It cannot artificially cause the highly resolved structure observed from the quiescent emission.

### Target position

We looked for 8.4 GHz auroral emission from near the photosphere of LSR J1835 + 3259 about its motion-corrected Gaia coordinates. No other radio sources were within the HSA primary beam. Curiously, we found that the motion-corrected Gaia coordinates for LSR J1835 + 3259 differ from the measured location of its aurorae in Epochs 1 and 2 (Extended Data Table 2) by approximately 51 mas and approximately 28 mas, respectively. In both epochs, the offset is primarily in the north–south direction.

We did not obtain geodetic calibrations, so all provided target astrometry is relative to the known position of our phase calibrator[64] (Extended Data Fig. 2). Phase referencing with the ICRF source J1835 + 3241 ties our target field to the phase calibrator's ICRF3 coordinates to within 1 mas accuracy[65], with an additional uncertainty of approximately 0.2 mas on the coordinates of the calibrator itself[66]. We confirm this by phase referencing our check source J1848 + 3219 in each epoch and find that it is approximately 1 mas offset in right ascension and no measurable offset in declination.

We also consider offsets between the known ICRF3 location[66] of our phase calibrator and its motion-corrected Gaia coordinates[23,56] (approximately 0.4 mas); and uncertainties in the J2016.0 Gaia reference coordinates (approximately 0.04 mas), parallax (approximately 0.06 mas) and accumulated proper motion (approximately 0.2 mas) for LSR J1835 + 3259 (refs. 23,56).

We estimate a total 3$\sigma$ positional uncertainty of order approximately 5 mas that cannot account for the position offset that we observe. Instead, it may point to a companion. For instance, VLBA observations of a different auroral ultracool dwarf, TVLM 513 – 46546, recently revealed an astrometric signal consistent with a Saturn-mass planet on a 221-day orbit[67]. Intriguingly, contemporaneously published CARMENES radial velocity measurements find an apparent amplitude of greater than or equal to 1,000 m s$^{-1}$ for LSR J1835 + 3259 but draw no conclusions on a hypothetical companion at present[68].

A companion inside the radio lobes of LSR J1835 + 3259 cannot plausibly cause the astrometric offset that we observe. Taking the greater of the two offsets as an astrometric signal[69] for a single companion with semimajor axis less than or equal to 14 $R_{UCD}$, we calculated a hypothetical companion mass greater than or equal to 3 $M_{\odot}$. Such a massive and hot companion is not evident in existing spectral[23,24] or radial velocity observations of LSR J1835 + 3259 (refs. 68,69). We, therefore, rule out a heretofore unresolved binary as an alternate explanation for the radio lobes that we observed. However, we note that existing observations cannot rule out terrestrial-sized companions within the radio lobes of LSR J1835 + 3259.

### Time-averaged flare luminosity

Early to mid-M dwarfs with high flare rates exhibit quiescent radio emissions attributed to frequent low-energy flaring[20]. This interpretation stems from a correlation[20] between their quiescent radio and quiescent X-ray luminosities, $L_R/L_X \approx 10^{-15.5}$ Hz$^{-1}$, seen in dozens of stars[15], which also applies to solar flares when extrapolated[70].

The quiescent radio emission from LSR J1835 + 3259 exceeds this correlation by over four orders of magnitude based on an X-ray upper limit[14,15] $L_X < 3.3 \times 10^{24}$ erg s$^{-1}$. Such a departure indicates that frequent low-energy flares are unlikely to cause its quiescent radio emission. However, the observed radio lobes around LSR J1835 + 3259 have a synchrotron cooling time[31] $\tau$ of order two months, where

$$\tau = (6.7 \times 10^8/B^2\gamma) \text{ seconds}$$

for a Lorentz factor $\gamma \approx 30$ and magnetic field $B \approx 2$ G in the source region (Main text). In contrast, X-ray flares on ultracool dwarfs decay on timescales of minutes to hours[71,72].

As such, we also consider whether large flares occurring every few days to months[36,37,73,74] can populate its magnetosphere with accelerated electrons that persist in producing detectable quiescent radio emissions even after X-ray flares decay. In such a scenario, LSR J1835 + 3259's time-averaged X-ray flare luminosity $\langle L_{X,\,flare}\rangle$ may be significantly higher than its quiescent X-ray luminosity $L_X$.

We examine whether $\langle L_{X,\,flare}\rangle$ can explain our target's high quiescent radio luminosity by restoring it to the radio versus X-ray flare correlation. To our knowledge, no published X-ray flare frequency distributions (FFDs) exist for ultracool dwarfs and no FFD is available at any wavelength for LSR J1835 + 3259. As a proxy, we used optical FFDs of ultracool dwarfs in a similar spectral type range to LSR J1835 + 3259. We then estimated $\langle L_{X,\,flare}\rangle$ by roughly scaling flare energies from optical to X-ray wavelengths[75].

The optical flare rate in the Kepler band of 12 ultracool dwarfs with spectral type M6–L0 (ref. 36) can be described in the form

$$N(>E) = 10^{\alpha}\left(\frac{E}{E_0}\right)^{-\beta}$$

with power-law index $\beta$ and where $10^{\alpha}$ gives the number $N$ of flares per hour that have energy $E$ greater than $E_0 = 10^{30}$ erg. Using the minimum and maximum flare energies $E_{\min}$ and $E_{\max}$ observed for each star, we integrated the flare energy released over time to obtain a time-averaged flare luminosity in erg s$^{-1}$ in the Kepler band of

$$\langle L_{Kepler,\,flare}\rangle = \left(\frac{\beta}{1-\beta}\right)\frac{10^{\alpha}}{3{,}600}\left[\left(\frac{E_{\max}}{E_0}\right)^{1-\beta} - \left(\frac{E_{\min}}{E_0}\right)^{1-\beta}\right]E_0$$

when the flare rate $10^{\alpha}$ is in h$^{-1}$. With their FFD parameters, we obtained $\langle L_{Kepler,\,flare}\rangle = 0.2$–$15 \times 10^{25}$ erg s$^{-1}$ for the various stars.

Translating from optical to X-ray flare frequency distributions is highly uncertain, due to the relatively few observations of X-ray flares on ultracool dwarfs[38,71,72,76,77]. Indeed, UV flare observations highlight the difficulty of translating FFDs between wavelengths: black-body models of optical flares can underpredict UV flare energies by a factor of approximately ten due to UV spectral lines[78] and UV flares are often undetected in optical due to poor photospheric contrast[79,80].

To minimize model-dependent difficulties, we used an optical-to-X-ray conversion factor grounded in observations. A study comparing solar and stellar flares[75] found that the X-ray flare energy is approximately 20% of the total (coronal plus photospheric) radiated flare energy in solar flares and 30% in a flare on the M dwarf AD Leo. They also calculated that approximately 16% of total radiated flare energy lies in the Kepler band when assuming a 9,000 K black body. We adopt their AD Leo ratio to estimate each object's time-averaged X-ray flare luminosity using $\langle L_{X,\,flare}\rangle/\langle L_{Kepler,\,flare}\rangle = 0.3/0.16 \approx 2$. This ratio implies that optical and X-ray flare energies are similar to within an order of magnitude, which agrees with simultaneous optical and X-ray flares observed on an ultracool dwarf (spectral type M8V)[71] and an early M dwarf[77,81].

Using the above factor of two conversion, we estimated that the rate of X-ray flare energy released in these ultracool dwarfs ranges from $\langle L_{X,\,flare}\rangle \approx 0.4$–$30 \times 10^{25}$ erg s$^{-1}$. The radio-X-ray correlation[20] thus predicts radio luminosities of order $10^9$–$10^{11}$ erg s$^{-1}$ Hz$^{-1}$. In contrast, the observed radio luminosity of LSR J1835 + 3259 is $2 \times 10^{13}$ erg s$^{-1}$ Hz$^{-1}$, calculated from its typical approximately 500 µJy 8.4 GHz radio flux density (Extended Data Table 1). Thus, even the most frequently flaring UCDs in this sample[36] have time-integrated flare luminosities two orders of magnitude too low to explain our target's quiescent radio emission. We therefore disfavour stellar flares as the primary

acceleration mechanism for producing LSR J1835 + 3259's quiescent radio luminosity.

This calculation rests on two assumptions: (1) on average, X-ray flare energy is of a similar order of magnitude to optical flare energy; and (2) the energy partition described by the radio versus X-ray correlation will apply to flares on ultracool dwarfs. Future multiwavelength flare studies on ultracool dwarfs testing these assumptions will further elucidate whether flare energy release can plausibly accelerate the radio-emitting electrons that populate ultracool dwarf radiation belts.

## Data availability

All radio data are available on the National Radio Astronomy Archive (data.nrao.edu) under VLBA Program BK222, PI Kao. Earth ephemerides for calculating parallax motion corrections are available from the NASA Jet Propulsion Laboratory Horizons online solar system data and ephemeris computation service (ssd.jpl.nasa.gov/horizons).

## Code availability

Raw radio data were processed with the Astronomical Image Processing System package, publicly accessible through the National Radio Astronomy Observatory (aips.nrao.edu). This work also made use of Astropy (www.astropy.org), a publicly available community-developed core Python package of tools and resources for astronomy[82–85].

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

**Acknowledgements** M.K. thanks J. S. Pineda and G. Hallinan for reviewing initial manuscript writing; M. Claussen for help in scheduling observations and coordinating with all arrays; E. Martin, R. Bowins-Rubin, B. Das, M. Knapp, R. Murray-Clay and J. Fortney for input; and K. Kelly, A. August and C. Kao. Support was provided by the National Aeronautics and Space Administraytion (NASA) through the NASA Hubble Fellowship grant HST-HF2-51411.001-A awarded by the Space Telescope Science Institute, which is operated by the Association of Universities for Research in Astronomy, Inc., for NASA, under contract NAS5-26555, and by the Heising-Simons Foundation through the 51 Pegasi b Fellowship grant 2021-2943. This work used observations made with the National Radio Astronomy Observatory's Very Long Baseline Array, Karl G. Jansky Very Large Array, the Green Bank Observatory's Robert C. Byrd Green Bank Telescope and the 100-m telescope of the Max-Planck-Institut für Radioastronomie at Effelsberg. The National Radio Astronomy Observatory and the Green Bank Observatory are facilities of the National Science Foundation operated under cooperative agreement by Associated Universities, Inc. This work also used the SIMBAD and VizieR databases operated at CDS, Strasbourg, France, and data from the European Space Agency mission Gaia (https://www.cosmos.esa.int/gaia), processed by the Gaia Data Processing and Analysis Consortium (DPAC, https://www.cosmos.esa.int/web/gaia/dpac/consortium). Funding for the DPAC has been provided by national institutions, in particular, the institutions participating in the Gaia Multilateral Agreement.

**Author contributions** This study was the result of equal efforts by M.K. and A.M. M.K. conceived and led the experimental design and proposal writing, coordinated and led observations, reduced all VLA-only pre-observing blocks, collaborated on HSA data analysis and led manuscript writing and revisions. A.M. worked closely with M.K. to design and coordinate observations, reduced all HSA science data presented in the article and contributed to manuscript writing. J.R.V. led the time-averaged flare luminosity calculation and contributed to the proposal design and overall manuscript writing. E.S. supported M.K. with funding during proposal writing, shaped the time-averaged flare luminosity calculation and contributed to overall manuscript revisions. All authors were co-investigators on the observing proposal that led to this work.

**Competing interests** The authors declare no competing interests.

**Additional information**
**Correspondence and requests for materials** should be addressed to Melodie M. Kao.

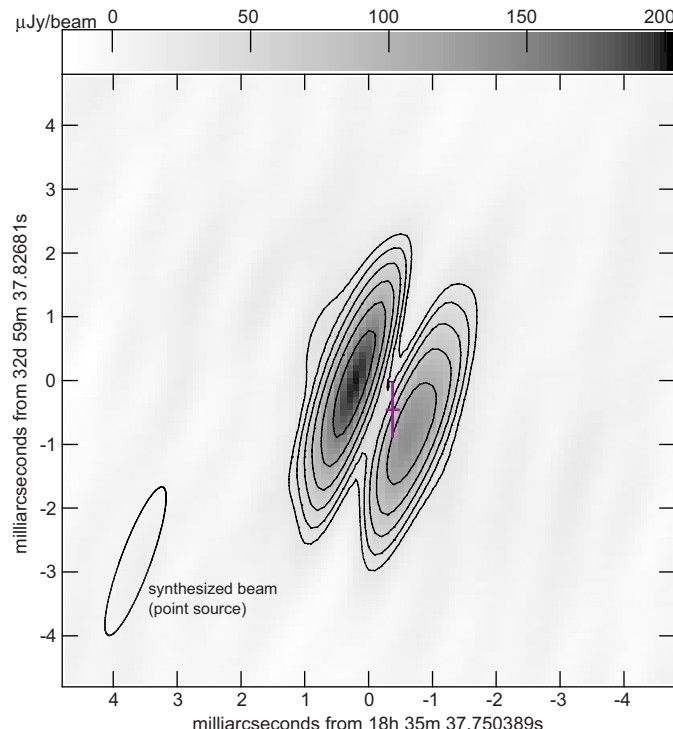

**Extended Data Fig. 1 | Simulated observation of the Epoch 2 model image using the Epoch 1 antenna configuration.** Epoch 1 science images are consistent with Epoch 2 when accounting for differences in antenna configurations. Measured lobe separation: $1.22 \pm 0.01$ mas ($13.56 \pm 0.64\, R_{\mathrm{UCD}}$).

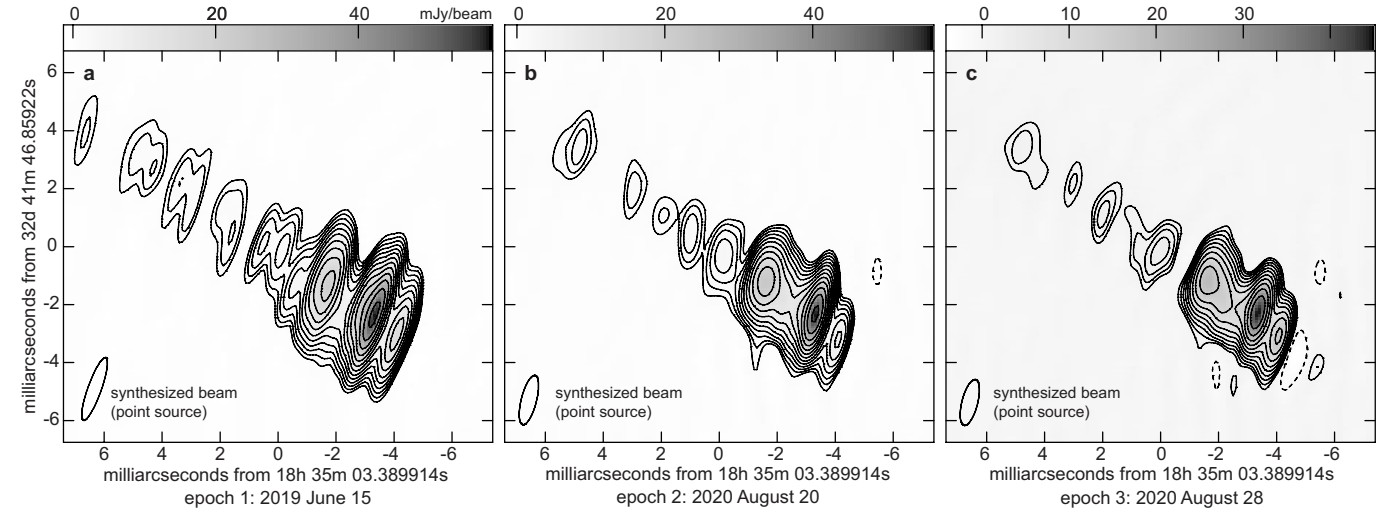

**Extended Data Fig. 2 | Phase calibrator J1835+3241 after self-calibration.** Contours denote $3\sigma_{rms} \times (-1, 1, \sqrt{2}, 2, 2\sqrt{2}, 4, 4\sqrt{2}, 8, 8\sqrt{2}, 16, 16\sqrt{2}, 32, 32\sqrt{2}, 64)$ increments.

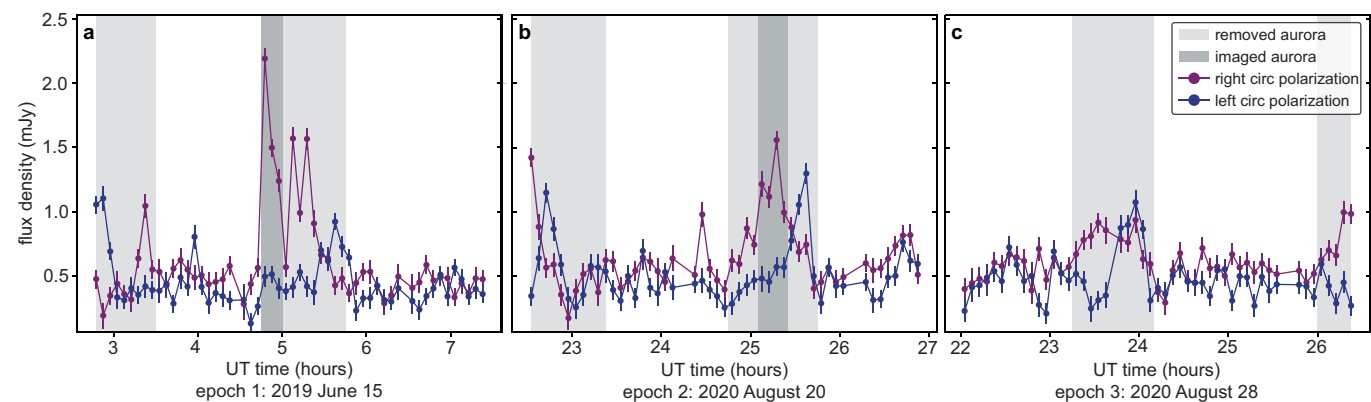

**Extended Data Fig. 3 | Timeseries for LSR J1835+3259.** Data for each epoch from only the phased-VLA are binned into 5-min intervals showing the right circularly polarized emission (magenta), left circularly polarized emission (blue), removed aurora (light grey), and imaged aurora (dark grey). Flux density errors are root mean squared errors. Calibrator and autophasing scans that extended for approximately 10 min and occurred every ~2 hr apart are evident and partially coincided with an aurora in Epoch 3 (panel c).

## Extended Data Table 1 | Properties of LSR J1835+3259

| Property | | | | ref. |
|---|---|---|---|---|
| R. A. | | 18 35 37.78772388 | (J2016.0) | 23 |
| Dec. | | +32 59 41.23377132 | (J2016.0) | 23 |
| Parallax | | 175.7930 ± 0.0468 | (mas) | 23 |
| Distance | | 5.6875 ± 0.002927 | (pc) | 23 |
| $\mu_\alpha \cos \delta$ | | −72.650 ± 0.047 | (mas yr$^{-1}$) | 23 |
| $\mu_\delta$ | | −755.146 ± 0.052 | (mas yr$^{-1}$) | 23 |
| Spectral type | | M8.5V | | 24 |
| Radius | | 1.07 ± 0.05 | ($R_J$) | 22 |
| Age | | ≥ 500 | (Myr) | 22 |
| M | | 77.28 ± 10.34 | ($M_J$) | 22 |
| Rot. period | | 2.845 ± 0.003 | (hr) | 85 |
| $i$ | | 90 | (°) | 22, 85 |
| $L_X$ | | < 3.3 × 10$^{24}$ | (erg s$^{-1}$) | 14 |
| $F_\nu$ * | 8.46 GHz | 464 ± 10 | (μJy) | 14 |
| | 8.44 GHz | 722 ± 15 | (μJy) | 25 |
| | — GHz | 525 ± 15 | (μJy) | 26 |
| | 97.5 GHz | 114 ± 18 | (μJy) | 27 |

*Stokes I quiescent radio flux densities reported in the literature. When available, listed observing frequencies are the centre of the reported observing band. Symbols used above: $\mu_\alpha \cos \delta$: proper motion in the right ascension (R. A.) direction, converted by declination (Dec., δ). $\mu_\delta$: proper motion in the declination direction. $i$: Inclination angle of rotation axis with respect to the line of sight. $L_x$: X-ray luminosity. $F_\nu$: Radio flux density.

**Extended Data Table 2 | Properties of quiescent and auroral radio emission from LSR J1835+3259**

| Epoch: Lobe[*] | $\sigma_{rms}$[†] | Centroid R. A.[‡§] | Centroid Dec.[‡§] | Peak $F_\nu$[§] | Integrated $F_\nu$[§] | Circ. Poln.[‖] | Major Axis[§] | Minor Axis[§] |
|---|---|---|---|---|---|---|---|---|
| | ($\mu$Jy beam$^{-1}$) | (hh mm ss) | (dd mm ss) | ($\mu$Jy/beam) | ($\mu$Jy) | (%) | (mas) | (mas) |
| 1: E | 13, 12 | 18 35 37.771464(1) | +32 59 38.77217(4) | 271 ± 12 | 342 ± 25 | ≤ 8.8 | 0.76 ± 0.10 | — |
| W | 13, 12 | 18 35 37.771401(2) | +32 59 38.77150(7) | 155 ± 12 | 201 ± 26 | ≤ 15.5 | 0.37 ± 0.17 | — |
| A[¶#] | 57, — | 18 35 37.771409(3) | +32 59 38.7714(13) | 908 ± 69 | 965 ± 12 | — | 1.19 ± 0.32 | — |
| 2: E | 12, 12 | 18 35 37.750407(3) | +32 59 37.82694(9) | 113 ± 11 | 235 ± 33 | ≤ 21.4 | 1.23 ± 0.20 | 0.58 ± 0.10 |
| W | 12, 12 | 18 35 37.750313(5) | +32 59 37.8258(10) | 98 ± 11 | 225 ± 35 | ≤ 25.0 | 1.42 ± 0.25 | 0.71 ± 0.12 |
| A[¶] | 43, — | 18 35 37.750354(4) | +32 59 37.8263(27) | 237 ± 40 | 537 ± 123 | — | 3.12 ± 0.63 | 0.40 ± 0.00 |
| 3: E | 13, 12 | 18 35 37.749165(5) | +32 59 37.7938(11) | 96 ± 12 | 189 ± 34 | ≤ 27.9 | 1.34 ± 0.26 | 0.66 ± 0.11 |
| W | 13, 12 | 18 35 37.749064(7) | +32 59 37.7927(18) | 65 ± 12 | 164 ± 40 | ≤ 44.0 | 1.64 ± 0.42 | 0.83 ± 0.19 |
| A | 43, — | — | — | ≤ 129 | — | — | — | — |

Minor axes in Epoch 1 are unresolved. Aurorae in Epoch 3 were not successfully imaged. [*]E: east quiescent lobe; W: west quiescent lobe; A: aurora. [†]Root mean squared errors. For east and west quiescent lobes, we obtained $\sigma_{rms}$ from images of the total (Stokes I) and circularly polarized (Stokes V) quiescent emission. For aurorae, we obtained $\sigma_{rms}$ from images of the right-circularly polarized emission. [‡]Uncertainties in the least significant digit are given in parentheses for coordinates, which are for midnight in International Atomic Time on the epoch date. [§]Fits and errors[63] from AIPS task "jmfit" using Stokes I (east and west quiescent lobes) or right-circularly polarized (aurora) emission. [‖]No circular polarization in the quiescent emission was detected in any epoch. We report 95% confidence upper limits for the absolute value of percent circular polarization in the peak emission calculated with $\sigma_{rms}$ from each epoch's Stokes V image. [¶]Synthesized beams for aurora: Epoch 1: 4.05×0.54 mas; Epoch 2: 2.06×0.55 mas. [#]Includes data from all antennas except for MK, SC, and GBT. Symbols and abbreviations used above: R.A.: right ascension. Dec.: declination. $F_\nu$: Radio flux density. Circ. Poln.: circular polarization.