## [Peer Review File · Nature]

Manuscript Title: Resolved imaging of an extrasolar radiation belt around an ultracool dwarf

Reviewer Comments & Author Rebuttals

Reviewer Reports on the Initial Version:

Referee #1 (Remarks to the Author):

This paper presents very convincing observational evidence for a radiation belt around an ultracool dwarf already known to emit periodic polarized bursts, probably of auroral origin. The images of the feature identified as a radiation belt over 3 epochs between June 2019 and August 2020 (the last 2 epochs just one week apart) are quite convincing. The measured flux density is consistent with theoretical/scaling law predictions, e.g. by Leto et al. (2015).

A secondary result, itself of importance in my opinion, is an image of a polarized auroral burst.

This work is a significant milestone in our discovery/understanding of stellar magnetospheres, and it undoubtedly deserves publication in Nature.

I have only relatively minor comments and suggestions below.

- It could be useful to give the star's name (LSR J1835+3259) in the title.
- The frequency of the observations could be given in the abstract.
- At top of p. 3,
"A search for companions around an auroral ultracool dwarf marginally resolved 8.5 GHz emission [27], which we propose hints at a possible radiation belt like Jupiter's."
is not a sentence. Please be more clear that it refers to the present paper.
- It seems to me that some numbers like 2316 ± 51 K or 77.28 ± 10.3 Mj or 18.47 ± 2.20 RUCD are given with too many (in)significant digits.
- Is there a reference describing the HSA ?
- In the paragraph first describing the results, at the bottom of p. 3,
"We find that quiescent radio emissions..."
mention to the linear polarization (expected from radiation belts) should be immediately mentioned. It would seem better to say "the first resolved radio imaging of an ultracool dwarf radiation belts" than "the first resolved radio imaging of an ultracool dwarf magnetosphere" which is not incorrect but slightly misleading.
- The following discussion of auroral bursts versus radiation belts emission that follows also insufficiently relies -in my opinion- on the polarization of the various components. Rather than the spatial coincidence of the various components and the simulation of observations, isn't it more

simple to rely on the polarization selection allowing to image separately the various components ?

- Table 2 and the corresponding discussion in the text are slightly confusing. The table and footnotes are too compact and thus difficult to decipher: what concerns Stokes I, Stokes V, each lobe, to what applies the mention "3 No circularly polarized emission was detected." should be clearly explained.

- I'm not sure that the remark

"Higher order magnetic fields fall off more rapidly in strength than dipole fields, resulting in higher harmonics."

is relevant, as radiation belt electrons (at planets) are essentially trapped in the dipolar field.

- Again, following the remark

"such high harmonics indicate synchrotron emission from very relativistic electrons, which cannot produce strong circular polarization"

could be followed by a mention on linear polarization.

The last sentence of p. 4 mentions the linear polarization measured at Jupiter, and adds that

"Similar measurements would confirm a synchrotron interpretation for LSR J1835+3259"

Does this mean that no linear polarization was measured in the reported observations ? Please state it clearly and explain why.

- Mention of results from ref. [29] in the last paragraph of p. 4, about the absence of circular polarization, does not add much to the discussion. The results obtained in the present paper seem self-consistent. Ref. [29] could just be cited at the end of the paragraph in support to conclusions drawn from the reported observations.

- The paragraph on p. 5 about flares and impulsive acceleration,

"Flares can impulsively accelerate electrons..."

that follows the estimation of the cooling time of belt electrons and the persistence of the observed features, appears as a complex and useless discussion.

The following mention that planetary radiation belts are long-lived structures is enough. This indeed requires a continuous refilling of the belts. Invoking adiabatic heating with ref. [4] makes sense, but the authors may want to indicate that the question of electron acceleration to MeV energies is no so simple and still very debated. For example, ref. [4] notes that

"Another theory that might explain the presence of MeV electrons at large distances is that they are derived from repeated adiabatic heating, where electrons run in a cycle of inward transport with heating at low latitudes and an energy-preserving outward transport at high latitudes (Fujimoto & Nishida, 1990; Nishida, 1976)."

and another good reference on the subject is:

Hill, Dessler & Goertz, Magnetospheric models, in "Physics of the Jovian Magnetosphere", ed. Dessler, Cambridge, 1983.

- At last line of p. 5, instead of "MHz radiation belts" rather mention "~100 MHz radiation belts". Girard et al. [ref. 44] observations are ≥ 127 MHz, and to my knowledge there are no observations below 74 MHz.

- What is "PA (°)" in Table 1 ? You probably mean "Major axes OF EACH LOBE are unresolved" ? Is that because they are parallel to the synthesized beam ?
- On Fig. 1 "Crosshairs indicate aurorae centroids" : can't you indicate the star's center as deduced from Gaia data ?
- At last line of p. 10: synchrotron emission is not circularly polarized. The formulation is ambiguous.
- On page 12, I don't understand the sentence:
"which was further exacerbated by a set of ~10 min extended calibration scans coinciding with one of the auroral bursts."
and the caption of Fig. 3
"Calibrator and autophasing scans that extended for ~10 minutes and occurred ~2 hours apart are evident in the data and partially coincided with an aurora in Epoch 3."
does not clarify the point.
- I must say that I find the term "excised" awful, and I encourage the use of an alternative such as "flagged out" or other.

Congratulations for a nice paper !

Philippe Zarka

Referee #2 (Remarks to the Author):

The authors present high angular resolution radio images (VLBI) of a nearby ultracool dwarf. This data reveal a double-lobed morphology that is persistent over 3 epochs spread over a year. The double-lobed morphology, its radio polarisation characteristics and the inferred approximate energy of the emitting electrons (15 MeV) is reminiscent of synchrotron emission from the so-called radiation belts around Jupiter, only it is several orders of magnitude brighter.

The main high-impact result is the claim that the observed radio emission is from a radiation belt which would make it the first extrasolar radiation belt. This claim is not water-tight but a reasonable supposition given the available evidence.

As such, my main criticism is regarding the claim that the emission is powered by an engine that is magnetospheric in origin (e.f. radial migration) rather than coronal/chromospheric (see major points below). I would like to see the authors buttress this claim better.

MAJOR POINTS

1) I find Table 1 to be confusing. Do the minor and major axes refer to the PSF beam and not the lobes themselves? Else I don't understand why "major axes are unresolved" while minor axes are

resolved. Without this specified, it is hard to interpret this table.

2) A double-lobe morphology is a consequence of electrons having small pitch angles being lost over a period of time leaving only electrons that have mirror points close to the magnetic equator (in addition to a projection effect). It cannot by itself prove that the electrons were accelerated in the magnetosphere (by radial migration for instance). For example, the magnetosphere of chemically peculiar stars are also expected to have this geometry (as the authors reference) but it is also possible that in these cases the electron acceleration happens via centrifugal breakout. As such, I would like to see the authors make a more forceful argument against the 15 MeV electrons being accelerated in numerous ongoing flares. Right now the flare-scenario is ruled out by saying that flares are impulsive. But the authors also show that the radiative cooling timescale is of the order of 2 months. So why can't intermittent flaring lead to a sustained population of synchrotron emitting electrons? Perhaps the authors can look into solar flare accelerated relativistic electrons as a template and see if they can arrive at an energetics-based contradiction in multi-wavelength flare observations (or flux densities) of the ultra-cool dwarf.

3) The non-central location of the burst in epoch 1 is attributed to the uv-plane sampling and sensitivity. Can the authors present a figure (perhaps in methods or suppl. info) that shows this and perhaps a short explanation of how missing uv-samples can shift the position of a source? It is important to demonstrate that this is an instrumental measurement effect and right now the reader has not much to go on.

4) The central burst in Figure 2 looks like it has two resolved components. The grayscale looks saturated so it is hard to say why but it does not look like a PSF lobe (they must be symmetrical). Is this due to temporal variability of the source? maybe the authors can do a simple simulation to show this?

5) I would naively expect that the rotation averaged images will have symmetry. But this is not the case. For example, Epoch 2 image has an asymmetric southern extent in its emission. Can the authors comment on why this is? i.e. provide a reasonable explanation.

MINOR POINTS

6) "Intriguingly, the quiescent radio luminosities of auroral ultracool dwarfs correlate well with their Balmer emission [11, 26] tracing auroral rather than the usual chromospheric activity" Can you check the reference provided for this statement. I suspect that the authors want to reference <https://iopscience.iop.org/article/10.3847/1538-4357/aa8596> here?

7) "emits at a narrow range of frequencies." Please rephrase. Synchrotron emission is broadband in general but is peaked at the critical frequency given in eqn (2)

8) "but that structure disappeared within hours" It is worth adding here that the double lobe structure of UV Ceti also appeared to be oriented along the rotation axis.

9) "and no infrared excess indicates the presence of a disk" Why will the lack of IR excess indicate the presence of a disk? Is this a typo?

10) "that rely on rapidly rotating magnetic dipole fields" I don't understand this. The authors suggest radial inward migration as a possible cause of acceleration (a'la Jupiter) but I am not aware that it is established that Jupiter's rotation forces the inward migration. Can the authors provide a reference for this?

11) Please use East and West in table 2 and elsewhere. Right and left may be ambiguous whereas East is understood by astronomers as direction of increasing RA.

Author Rebuttals to Initial Comments:

Referee #1 (Remarks to the Author)

This paper presents very convincing observational evidence for a radiation belt around an ultracool dwarf already known to emit periodic polarized bursts, probably of auroral origin. The images of the feature identified as a radiation belt over 3 epochs between June 2019 and August 2020 (the last 2 epochs just one week apart) are quite convincing. The measured flux density is consistent with theoretical/scaling law predictions, e.g. by Leto et al. (2015).

A secondary result, itself of importance in my opinion, is an image of a polarized auroral burst.

This work is a significant milestone in our discovery/understanding of stellar magnetospheres, and it undoubtedly deserves publication in Nature. I have only relatively minor comments and suggestions below.

1. It could be useful to give the star's name (LSR J1835+3259) in the title.

Thank you for this suggestion. We have decided not to implement it given Nature's broader readership.

2. The frequency of the observations could be given in the abstract.

Done.

3. At top of p. 3, "A search for companions around an auroral ultracool dwarf marginally resolved 8.5 GHz emission [27], which we propose hints at a possible radiation belt like Jupiter's." is not a sentence. Please be more clear that it refers to the present paper.

We have clarified this sentence.

4. It seems to me that some numbers like 2316 ± 51 K or 77.28 ± 10.3 Mj or 18.47 ± 2.20 RUCD are given with too many (in)significant digits.

Thank you for pointing out your concern. For parameters quoted from published papers (e.g. effective temperature, mass), we defer to the published significant digits. For the measured centroid separations, we chose significant digits based on our imaging resolution. Each synthesized beam and pixel are ~ 0.5 mas and 0.1 mas, respectively. For context, an ultracool dwarf radius is ~ 0.9 mas. As such, we provided one additional significant digit to give a measure of the uncertainty. Therefore, we elect not to reduce the number of significant digits in our reported centroid separations or lobe diameters.

5. Is there a reference describing the HSA?

To our knowledge, no. It is not a purpose-built VLBI array. Instead, it combines several existing telescopes and/or arrays, most of which fall under the management of the National Radio Astronomy Observatory.

6. In the paragraph first describing the results, at the bottom of p. 3, "We find that quiescent radio emissions..." mention to the linear polarization (expected from radiation belts) should be immediately mentioned.

The presented data don't include linear polarization, so we cannot make a statement about observed linear polarization. We make this more clear in our linear polarization discussion later in the manuscript: "Synchrotron emission instead produces linear polarization, which has been observed at the 20% level for the Jovian radiation belts. Our observations do not include linear polarization calibrations, calling for future such measurements to confirm synchrotron emission from LSR J1835+3259."

7. It would seem better to say "the first resolved radio imaging of an ultracool dwarf radiation belts" than "the first resolved radio imaging of an ultracool dwarf magnetosphere" which is not incorrect but slightly misleading.

We revised this statement to say: "These data constitute the first resolved imaging of plasma captured in the magnetosphere of a planet-sized object outside of our Solar System." We felt that it would be premature to call it a radiation belt until we took the reader through our line of reasoning. If the Editor would like us to change our revised wording to the referee's suggested wording, we are happy to do so.

8. The following discussion of auroral bursts versus radiation belts emission that follows also insufficiently relies -in my opinion- on the polarization of the various components. Rather than the spatial coincidence of the various components and the simulation of observations, isn't it more simple to rely on the polarization selection allowing to image separately the various components?

We included the spatial information as groundwork for the line of reasoning concluding that the quiescent emission is synchrotron emission from MeV electrons. However, the referee makes an excellent point here and we elaborate on how we use circular polarization in the Methods section. Essentially, we use the circular polarization to double check the imaging of the aurorae vs. the radiation belts.

9. Table 2 and the corresponding discussion in the text are slightly confusing. The table and footnotes are too compact and thus difficult to decipher: what concerns Stokes I, Stokes V, each lobe, to what applies the mention "3 No circularly polarized emission was detected." should be clearly explained.

We have revised the notes in Table 2 (now condensed to Extended Data Table 2) and the text.

10. I'm not sure that the remark "Higher order magnetic fields fall off more rapidly in strength than dipole fields, resulting in higher harmonics." is relevant, as radiation belt electrons (at planets) are essentially trapped in the dipolar field.

We are uncomfortable with removing the existing statement, as we do not want to assume that radiation belts may only exist in dipole fields given the very limited existing dataset of solar system planets and now LSR J1835+3259. Furthermore, our quantitative line of reasoning shows that our results stand regardless of the actual magnetic field topology, which we believe strengthens our interpretation. Until this work, no observations definitively established that our target had a dipole magnetic field. In fact, magnetic field modeling from line broadening suggests that our target also has strong higher-order fields, which we believe need to be addressed in our interpretation. As such, we elected to also write for a skeptical audience, such as those who may view the quiescent emission that we observe from the lens of stellar flaring.

11. Again, following the remark "such high harmonics indicate synchrotron emission from very relativistic electrons, which cannot produce strong circular polarization" could be followed by a mention on linear polarization.

Thank you for this suggestion. We considered this, but doing so interrupts the flow of discussion so we have decided not to implement it.

12. The last sentence of p. 4 mentions the linear polarization measured at Jupiter, and adds that "Similar measurements would confirm a synchrotron interpretation for LSR J1835+3259." Does this mean that no linear polarization was measured in the reported observations? Please state it clearly and explain why.

Done.

13. Mention of results from ref. [29] in the last paragraph of p. 4, about the absence of circular polarization, does not add much to the discussion. The results obtained in the present paper seem self-consistent. Ref. [29] could just be cited at the end of the paragraph in support to conclusions drawn from the reported observations.

We have rewritten this paragraph to emphasize our own observations and their self-consistency:

"Instead, such high harmonics indicate synchrotron emission from very relativistic electrons, which cannot produce strong circular polarization. Indeed, we do not detect circular polarization in its resolved radio lobes in any epoch (Table~3; see also Methods). For the less resolved and brighter quiescent emission in Epoch 1, our noise floor gives a 95% confidence upper limit of $\leq 8.8\%$ and $\leq 15.5\%$ circular polarization in the east and west lobes, respectively."

14. The paragraph on p. 5 about flares and impulsive acceleration, "Flares can impulsively accelerate electrons..." that follows the estimation of the cooling time of belt electrons and the persistence of the observed features, appears as a complex and useless discussion. The following mention that planetary radiation belts are long-lived structures is enough. This indeed requires a continuous refilling of the belts.

We respectfully disagree but are pleased with the referee's enthusiasm for the radiation belt interpretation of our data. We included this discussion because of historical interpretations that the quiescent emission was from flare-accelerated electrons, which need to be addressed (see also Referee 2's Major Comment #2). However, we've re-written this paragraph to improve clarity.

15. Invoking adiabatic heating with ref. [4] makes sense, but the authors may want to indicate that the question of electron acceleration to MeV energies is not so simple and still very debated. For example, ref. [4] notes that Another theory that might explain the presence of MeV electrons at large distances is that they are derived from repeated adiabatic heating, where electrons run in a cycle of inward transport with heating at low latitudes and an energy-preserving outward transport at high latitudes (Fujimoto & Nishida, 1990; Nishida, 1976)." and another good reference on the subject is: Hill, Dessler & Goertz, Magnetospheric models, in "Physics of the Jovian Magnetosphere", ed. Dessler, Cambridge, 1983.

We have softened our language to reflect that that adiabatic diffusion is but one theory that has borne out in recent observations of Jupiter: "Radiation belts around Solar System planets offer alternative acceleration mechanisms and a compelling analogy for interpreting LSR J1835+3259's double-lobed quiescent radio emission. In contrast to stellar flares, centrifugally outflowing plasma accelerates while maintaining co-rotation with Jupiter's magnetosphere⁴, stretching and triggering reconnection in its global magnetic field at large distances⁴². Rotationally-driven currents spanning Jupiter's magnetosphere⁴ and powering its main aurora⁴³ can also accelerate radiation belt electrons. These processes effectively tap Jupiter's large reservoir of rotational energy. Finally, to reach observed MeV energies, electrons undergo adiabatic heating as they encounter stronger magnetic fields during slow inward radial diffusion⁴."

16. At last line of p. 5, instead of "MHz radiation belts" rather mention "~100 MHz radiation belts". Girard et al. [ref. 44] observations are ≥ 127 MHz, and to my knowledge there are no observations below 74 MHz.

Done.

17. What is "PA ($^{\circ}$)" in Table 1? You probably mean "Major axes OF EACH LOBE are unresolved" ? Is that because they are parallel to the synthesized beam ?

We have revised the tables to remove the Position Angle (PA) since we don't use this information in the discussion (all relevant data for an interested reader to calculate the position angles between the lobes are supplied in the tables). We also revised the text to clarify that the major axes of each lobe are unresolved at the suggestion of the referee: "Lobe diameters and errors obtained by fitting two freely-floating elliptical Gaussians to each epoch image. Fitted minor axes for each lobe are resolved except for in Epoch 1 due to missing antennas. Major axes for each lobe are unresolved in fits."

18. On Fig. 1 "Crosshairs indicate aurorae centroids": can't you indicate the star's center as deduced from Gaia data?

Unfortunately not. 3-sigma positional errors for LSR J1835+3259 exceed our image sizes when we include uncertainties on its reference coordinates, proper motion, parallax, the coordinates of the phase calibrator which tie our target field to the ICRF, and more. Moreover, there is a positional offset compared to the Gaia coordinates that these uncertainties cannot explain. We detail this offset and its possible implications in Methods: Target Position.

19. At last line of p. 10: synchrotron emission is not circularly polarized. The formulation is ambiguous.

We have clarified the text: "Circular polarization, which is the difference in the right and left circularly polarized data, can distinguish between and characterize electron cyclotron maser (~100% circular polarization), gyrosynchrotron (up to tens of percent), and synchrotron emissions (minimal)".

19. On page 12, I don't understand the sentence: "which was further exacerbated by a set of ~10 min extended calibration scans coinciding with one of the auroral bursts." and the caption of Fig. 3. "Calibrator and autophasing scans that extended for ~10 minutes and occurred ~2 hours apart are evident in the data and partially coincided with an aurora in Epoch 3." does not clarify the point.

We have revised the text to state: "In Epoch 3, auroral bursts were too faint to confidently image. A set of extended calibration scans coincided with one of the bursts (Extended Data Figure 3). As a result, no data were obtained on-target during its peak brightness, suppressing the contrast between auroral flux density and imaging rms noise."

20. I must say that I find the term "excised" awful, and I encourage the use of an alternative such as "flagged out" or other.

Noted! We have substituted "excised" with "removed".

Congratulations for a nice paper!
Philippe Zarka

Referee #2 (Remarks to the Author)

The authors present high angular resolution radio images (VLBI) of a nearby ultracool dwarf. This data reveal a double-lobed morphology that is persistent over 3 epochs spread over a year. The double-lobed morphology, its radio polarisation characteristics and the inferred approximate energy of the emitting electrons (15 MeV) is reminiscent of synchrotron emission from the so-called radiation belts around Jupiter, only it is several orders of magnitude brighter.

The main high-impact result is the claim that the observed radio emission is from a radiation belt which would make it the first extrasolar radiation belt. This claim is not water-tight but a reasonable supposition given the available evidence.

As such, my main criticism is regarding the claim that the emission is powered by an engine that is magnetospheric in origin (e.g. radial migration) rather than coronal/chromospheric (see major points below). I would like to see the authors buttress this claim better.

MAJOR POINTS

1. I find Table 1 to be confusing. Do the minor and major axes refer to the PSF beam and not the lobes themselves? Else I don't understand why "major axes are unresolved" while minor axes are resolved. Without this specified, it is hard to interpret this table.

We have clarified the Table 1 notes to state: "Lobe diameters and errors obtained by fitting two freely-floating elliptical Gaussians to each epoch image. Fitted minor axes for each lobe are resolved except for in Epoch 1 due to missing antennas. Major axes for each lobe are unresolved in fits."

2. A double-lobe morphology is a consequence of electrons having small pitch angles being lost over a period of time leaving only electrons that have mirror points close to the magnetic equator (in addition to a projection effect). It cannot by itself prove that the electrons were accelerated in the magnetosphere (by radial migration for instance). For example, the magnetosphere of chemically peculiar stars are also expected to have this geometry (as the authors reference) but it is also possible that in these cases the electron acceleration happens via centrifugal breakout. As such, I would like to see the authors make a more forceful argument against the 15 MeV electrons being accelerated in numerous ongoing flares.

Right now the flare-scenario is ruled out by saying that flares are impulsive. But the authors also show that the radiative cooling timescale is of the order of 2 months. So why can't intermittent flaring lead to a sustained population of synchrotron emitting electrons? Perhaps the authors can look into solar flare accelerated relativistic electrons as a template and see if they can arrive at an energetics-based contradiction in multi-wavelength flare observations (or flux densities) of the ultra-cool dwarf.

Many thanks to the referee for encouraging stronger arguments against the 15 MeV electrons being accelerated by a series of flares, a deeper discussion about flares and radiation belts in general, and for the excellent suggestion of considering an energetics-based contradiction. By investigating the referee's recommendations, we are able to significantly strengthen our narrative for the interpretation of our results.

The referee's comments also led us to consider how we define a radiation belt: is it the morphology or the acceleration mechanism? In paragraph 4 on page 3, we define a radiation belt in terms of morphology: "long-lived relativistic electron populations confined in a global magnetic dipole field". This morphological definition acknowledges the diversity of acceleration mechanisms present in Solar System radiation belts, which work together in multistage heating processes to accelerate radiation belt particles to their observed MeV energies. Doing so has the added benefit of providing a means of unifying stellar and planetary magnetic activity phenomena by not precluding stellar-like flares as a possible *source* of electrons (or even an earlier stage in an ultracool dwarf radiation belt acceleration mechanisms chain), even though they do not accelerate Jupiter's radiation belt electrons in-situ. In the concluding paragraph of the Main Text, we highlight that ultracool dwarf magnetic activity occupies an interesting space in the continuum between stars and planets – they appear capable of both flare activity akin to what is observed on low mass stars and also planet-like magnetospheric phenomena (aurorae, radiation belts). Regardless of the acceleration mechanism, our HSA imaging supports this morphology of a toroidal distribution of relativistic electrons.

However, with the energetics-based flare calculations that we added at the Referee's suggestion (described in the next paragraph), we are more confidently able to disfavor stellar-like flares (caused by small-scale – rather than global – magnetic reconnection driven by photospheric motions) as the *primary* acceleration mechanism for the MeV electrons imaged by our data. This then points to rotationally-driven acceleration mechanisms such as those seen in solar system planets or in the centrifugal breakout model as a more compelling alternative. In response to the referee's point about centrifugal breakout, we have also added a discussion in the Main Text raising this as a possibility.

Finally, we thank the referee for the suggestion of considering an energetics-based contradiction tied to solar flares. We have implemented this suggestion in the Methods and updated the Main Text to reflect this. To do so, we use Kepler-band flare frequency distributions for UCDs in the M6-L0 range (since a flare frequency distribution is not available for our target) to calculate their time-averaged flare luminosity. We then use scale factors between Kepler & X-ray flare energies from Osten & Wolk 2015 to translate this to a time-averaged flare X-ray luminosity. Using the Gudel-Benz relation that correlates X-ray and radio luminosity (and that applies to solar radio/X-ray flares and quiescent emission from magnetically active M dwarf coronae), we predict that the time-averaged radio flaring luminosity should be $\sim 10^9 - 10^{11}$ erg/s/Hz: two to four orders of magnitude lower than the observed quiescent radio emission from this target. Therefore, even though our target's long synchrotron cooling time provides an avenue for flare-accelerated electrons to persist long after a flare, the time-averaged flare energy budget fails to predict the quiescent radio luminosity of the target.

3. The non-central location of the burst in epoch 1 is attributed to the uv-plane sampling and sensitivity. Can the authors present a figure (perhaps in methods or suppl. info) that shows this and perhaps a short explanation of how missing uv-samples can shift the position of a source? It is important to demonstrate that this is an instrumental measurement effect and right now the reader has not much to go on.

We have added the figure of this simulated observation in the Extended Data and included a short discussion: "In Epoch 1, missing antennas (Table 1) significantly reduce sensitivity on shorter baselines relevant for detecting extended emission, causing the lobes to appear more compact and aurora to coincide with the west lobe (Figure 1a)."

4. The central burst in Figure 2 looks like it has two resolved components. The grayscale looks saturated so it is hard to say why but it does not look like a PSF lobe (they must be symmetrical). Is this due to temporal variability of the source? Maybe the authors can do a simple simulation to show this?

The short duration of the imaged aurora results in very limited baseline phase space (uv distance) coverage. This precludes robust modeling of the baseline-dependent flux density data to determine if the auroral burst in Epoch 2 is resolved with substructure. Attempting to do so would be an overinterpretation of the data. Given the data available, we are not comfortable conclusively stating that it is or is not resolved with substructure. However, we have added an additional discussion in the Methods section regarding this issue:

“In Epoch 2, the auroral burst is morphologically distinct from the quiescent radio lobes (Figure 2) and consistent with both being unresolved or marginally resolved along approximately the east-west direction. It has a minor axis of ~ 0.4 mas compared to a 2.06×0.55 mas synthesized beam (Table 4).

Even with our close-proximity phase calibrator, phase errors likely remain. These can introduce spurious substructure on length scales smaller than the synthesized beam for transient emission. Consequently, we cannot conclude whether the aurora has any physical substructure. This effect averages out over time as phase errors vary and is important only for assessing marginally resolved emission. It cannot artificially cause the highly resolved structure observed from the quiescent emission.”

5. I would naively expect that the rotation averaged images will have symmetry. But this is not the case. For example, Epoch 2 image has an asymmetric southern extent in its emission. Can the authors comment on why this is? i.e. provide a reasonable explanation.

We do not recommend attaching much significance to this apparent asymmetry given the challenges of VLBI imaging. It appears asymmetric primarily due to the notch in the southern portion of the western lobe, but this can be an artifact of image reconstruction. Providing an explanation beyond this would be an overinterpretation of the image given the presently available data.

MINOR POINTS

6. “that rely on rapidly rotating magnetic dipole fields” I don't understand this. The authors suggest radial inward migration as a possible cause of acceleration (a'la Jupiter) but I am not aware that it is established that Jupiter's rotation forces the inward migration. Can the authors provide a reference for this?

We have expanded the discussion to make the connection to rotational energy more explicit. See our response to Point 2 by this same referee.

7. “emits at a narrow range of frequencies.” Please rephrase. Synchrotron emission is broadband in general but is peaked at the critical frequency given in eqn (2)

We have rephrased to: “For synchrotron emission, we can estimate electron energies because each electron emits most of its power near its critical frequency $\nu_{\text{crit}} \approx (3/2) \gamma^2 \nu_c \sin \alpha$ ”

8) “Intriguingly, the quiescent radio luminosities of auroral ultracool dwarfs correlate well with their Balmer emission [11, 26] tracing auroral rather than the usual chromospheric activity” Can you check the reference provides for this statement. I suspect that the authors want to reference <https://iopscience.iop.org/article/10.3847/1538-4357/aa8596> here?

Thanks for catching this omission. We included the Pineda+ 2017 reference but kept the Kao+ 2016 reference since the efficacy of selecting brown dwarfs with Balmer line emission was one of their central results and a key piece of evidence arguing for aurorae on cold brown dwarfs.

9) “but that structure disappeared within hours” It is worth adding here that the double lobe structure of UV Ceti also appeared to be oriented along the rotation axis.

Done.

10) “and no infrared excess indicates the presence of a disk” Why will the lack of IR excess indicate the presence of a disk? Is this a typo?

Thank you for pointing out this ambiguity in our writing. We have revised the text to clarify: “Indeed, although the young inferred age does not exceed typical M dwarf disk dissipation timescales LSR~J1835+3259 does not exhibit infrared excess indicative of a disk”

11) Please use East and West in table 2 and elsewhere. Right and left may be ambiguous whereas East is understood by astronomers are direction of increasing RA.

Done!